# FlowBack: A flow-matching approach for generative backmapping of macromolecules

**Michael S. Jones** [1]  **Smayan Khanna** [1]  **Andrew L. Ferguson** [1]

## Abstract

Coarse-grained models have become ubiquitous in biomolecular modeling tasks aimed at studying slow dynamical processes such as protein folding and DNA hybridization. Although these models considerably accelerate sampling, it remains challenging to recover an ensemble of all-atom structures corresponding to coarse-grained simulations. In this work, we introduce a generative approach called FlowBack that uses a flow-matching objective to map samples from a coarse-grained prior distribution to an all-atom data distribution. We construct our prior distribution to be amenable to any coarse-grained map and any type of macromolecule, and we find that generated structures are more robust and contain less steric clashes than those generated by previous approaches. We train a protein-specific model on structures from the Protein Data Bank which achieve state-of-the-art results on bond quality and on clash score. Furthermore, we train a model on DNA-protein data which achieves excellent reconstruction and generative capabilities on complexes from the PDB as well as on coarse-grained simulations of DNA-protein binding.

## 1. Introduction

For decades, coarse-grained (CG) force-fields have expanded the time and length scales accessible to molecular dynamics simulations. By reducing the simulated degrees of freedom and smoothing the underlying free energy landscape, these simulation techniques can directly provide insight into slow processes and rare events such as protein folding and DNA hybridization (Clementi, 2008; Noid, 2013; Saunders & Voth, 2013; Kmiecik et al., 2016; Mohr et al., 2022; Shmilovich et al., 2020). However, the

[1]UChicago, Pritzker School of Molecular Engineering, University of Chicago, Chicago, Illinois 60637, United States. Correspondence to: Andrew Ferguson <andrewferguson@uchicago.edu>.

*Accepted at the 1st Machine Learning for Life and Material Sciences Workshop at ICML 2024.* Copyright 2024 by the author(s).

finer-grained, all-atom (AA) details of CG simulations are often of interest to i) obtain structure and dynamics that expose molecular mechanisms ii) access physical observables contingent on the AA coordinates for comparison with experimental data (e.g., X-ray scattering) or iii) determine the validity of the CG force-fields (Badaczewska-Dawid et al., 2020; Nishimura et al., 2024).

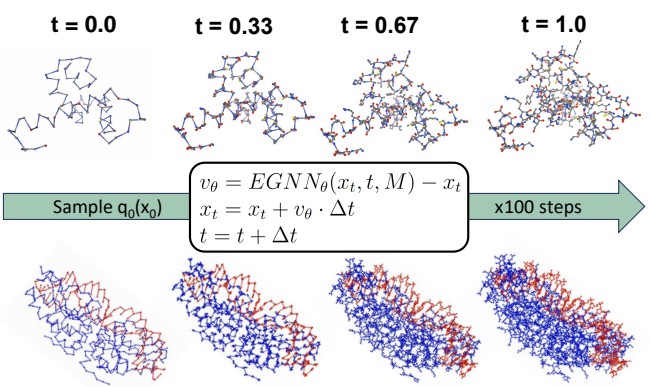

$$v_\theta = EGNN_\theta(x_t, t, M) - x_t$$
$$x_t = x_t + v_\theta \cdot \Delta t$$
$$t = t + \Delta t$$

*Figure 1.* FlowBack enables accurate, transferable, and efficient flow-based backmapping from coarse-grained to all-atom models of proteins (top) and protein-DNA complexes (bottom).

Backmapping is the process of recovering AA configurations from CG data, and numerous methods have been developed for this purpose ranging from rules-based approaches to data-driven ones. The former category generally performs geometrically guided initialization or structures from a fragment library then performs structural refinement and/or energy minimization to resolve problematic interactions (Lombardi et al., 2016; Wassenaar et al., 2014; Gopal et al., 2010; Brocos et al., 2012; Machado & Pantano, 2016; Rotkiewicz & Skolnick, 2008). These approaches work well in a number of cases, but are typically deterministic, and therefore fail to capture the ensemble of all-atom configurations compatible with a single coarse-grained structure, and may suffer from time-intensive energy minimization. Increasingly, data-driven approaches have been applied to the problem of molecular backmapping. Most early approaches were based on Generative Adversarial Networks (GANs) or Variational Autoencoders (VAEs) architectures and were

non-transferable across different protein sequences (Stieffenhofer et al., 2021; 2020; An & Deshmukh, 2020; Wang et al., 2022; Shmilovich et al., 2022)

Recently, several generative approaches were developed for transferable backmapping of protein traces (Yang & Gómez-Bombarelli, 2023; Jones et al., 2023; Liu et al., 2023; Chennakesavalu & Rotskoff, 2024). These approaches variously employ a CG-conditioned VAE, mapping tripeptide conformations via a transformer, and denoising diffusion models. Although these works have shown great promise in producing generative samples, they have limitations associated with narrow training sets (Liu et al., 2023; Yang & Gómez-Bombarelli, 2023), exhaustive cost of generating fragment ensembles (Chennakesavalu & Rotskoff, 2024), mode collapse (Yang & Gómez-Bombarelli, 2023), and long inference time (Jones et al., 2023; Liu et al., 2023).

In this work, we present FlowBack as a flow-based approach to generative backmapping. We leverage the recently developed flow-matching objective (Lipman et al., 2022; Albergo & Vanden-Eijnden, 2022; Liu et al., 2022) to accurately, efficiently, and transferably generate physically plausible and statistically meaningful AA structures from CG simulations. Like diffusion-based approaches, flow-matching learns to transform a noisy prior into a target data distribution. Unlike diffusion models, the prior can be much more flexible and bespoke to a given molecular structure. Furthermore, the linear nature of the flow-based interpolant and deterministic integrator often leads to more efficient training and inference than diffusion models (Song et al., 2024; Liu et al., 2022). These properties have already been leveraged to generate novel protein backbones (Yim et al., 2023; Bose et al., 2023), protein configurational ensembles (Jing et al., 2024), receptor-targeted peptide design (Lin et al., 2024), and small molecules (Song et al., 2024) with greater efficiency than analogous diffusion-based approaches (Trippe et al., 2022; Wu et al., 2022; Watson et al., 2022)). We use this framework to treat backmapping as a super-resolution problem, where the positions of all AA atoms are initialized with some noise surrounding their respective CG beads. This serves as an informative and roto-translationally invariant prior that can be generalized to arbitrary coarse-graining mappings and other classes of biomolecules.

We train a protein-specific model on structures up to 1000-residues from the Protein Data Bank (PDB). We evaluate this model on benchmarks that include static PDB structures, AA trajectories of fast-folding proteins, and CG trajectories generated by a machine-learned (ML) forcefield, and we demonstrate superior performance compared to previous models. Furthermore, we demonstrate the flexibility of our CG prior by training a second model to backmap CG DNA-protein structures. We train this model on a much smaller subset of structures from the PDB but still observe excellent reconstruction of both static structures and CG trajectories.

## 2. Methods

### 2.1. Flow-matching

Inspired by continuous normalizing flows (CNF)(Chen et al., 2018), the flow-matching framework defines a deterministic vector field $v_\theta(t, x)$ that integrates some prior distribution $q_0$ to a distribution approximating the training data $q_1$ where $t$ is the integration time and $x$ is an intermediate sample (Lipman et al., 2022; Albergo & Vanden-Eijnden, 2022; Liu et al., 2022). Training is conducted by using a simulation-free loss function that can be defined by regressing against the target vector field $u_t(x)$ and produces the data probability distribution $p_t(x)$,

$$\mathcal{L}(\theta) = \mathbb{E}_{t, x \sim p_t(x)} \left[ ||v_\theta(t, x) - u_t(x)||^2 \right]. \quad (1)$$

In practice $u_t$ is challenging to calculate, but it has been shown (Lipman et al., 2022) that it is possible to instead regress against the conditional vector field $u_t(x|x_1)$ which produces the probability path $p_t(x|x_1)$ given training samples from the target data distribution $q(x_1)$,

$$\mathcal{L}(\theta) = \mathbb{E}_{t, x_1 \sim q(x_1), x \sim p_t(x|x_1)} \left[ ||v_\theta(t, x) - u_t(x|x_1)||^2 \right]. \quad (2)$$

It was further shown (Albergo & Vanden-Eijnden, 2022; Tong et al., 2023) that the vector field can be defined from an arbitrary source distribution by sampling from the joint distribution $q_0(x_0)q_1(x_1)$ where $q(x_0)$ is our prior distribution. In the context of molecular backmapping, given an AA training structure $x_1$ and a mapping $M$ of AA atoms to CG beads, we construct our prior distribution by normally distributing each AA atom around its cognate CG bead as $x_0 \sim \mathcal{N}(x_1[M], \sigma_p^2\mathbf{I})$, where $x_1[M]$ indicates the CG mapping of the AA structure. The parameter $\sigma_p$ determines how tightly the AA beads are distributed, and the distribution of structures serves as a physically informed prior $q_0(x_0)$. Given samples $x_1, x_0$ from our training data and prior distributions, we can define our relative vector field $u_t$ as the difference between these structures $x_1 - x_0$ and obtain a linearly interpolated and noised structure $x_t$ at time point $t$ along the flow,

$$\mu_t = tx_1 + (1 - t)x_0$$
$$x_t \sim \mathcal{N}(\mu_t, \sigma_t^2\mathbf{I}). \quad (3)$$

### 2.2. Training

To train the model, we sample the vector field at various times and generate an interpolated and noised structure $x_t$.

We pass this structure to an Equivariant Graph Neural Network (EGNN) and compute a learned vector field $v_\theta$ as the difference between the output and input structures. We adopt this strategy from previous approaches (Hoogeboom et al., 2022; Igashov et al., 2022) which show that EGNNs are better at predicting configurations rather than noise or fields directly. Details of the EGNN architecture can be found in Appendix A4.2. We pass the network our CG map $M$ to ensure that CG beads remain constant in the predicted structure. We estimate the vector by finding the Euclidean difference between the interpolated and predicted structures,

$$v_\theta = EGNN(x_t, t, M) - x_t. \tag{4}$$

This learned vector field can then be regressed against the reference field $u_t$ to form our flow-matching loss. The complete training loop is presented in Algorithm 1.

---

**Algorithm 1** Training Loop
---
1: **Input:** Training data distribution $q_1$, masked prior $q_0$, prior noise $\sigma_p$, flow-matching noise $\sigma_t$, model $EGNN_\theta$, coarse-grained mask $M$
2: **for** $epoch = 1$ **to** $N$ **do**
3:     $x_1 \sim q_1(x_1)$
4:     $x_0 \sim q_0(x_0) \leftarrow \mathcal{N}(x_1[M], \sigma_p^2\mathbf{I})$
5:     $t \sim \mathcal{U}(0, 1)$
6:     $\mu_t \leftarrow tx_1 + (1-t)x_0$
7:     $x_t \sim \mathcal{N}(\mu_t, \sigma_t^2\mathbf{I})$
8:     $x_t[M] \leftarrow \mu_t[M]$
9:     $v_\theta \leftarrow EGNN_\theta(x_t, t, M) - x_t$
10:     $u_t \leftarrow x_1 - x_0$
11:     Take gradient step $\nabla_\theta \|v_\theta - u_t\|^2$
12: **end for**
13: **return** $EGNN_\theta$
---

Details on optimization of $\sigma_p$ and $\sigma_t$ are in Appendix A4.1.

## 2.3. Inference

Given a vector field $v_\theta(x, t)$ parameterized by our trained EGNN, inference proceeds by i) sampling an initial structure $x_0$ from the coarse-grained prior $q_0(x_0)$, ii) updating $x_t$ and $t$ via Euler integration, and iii) repeating this process for $N$ time steps. To ensure an exact correspondence between the CG and AA configurations, we enforce that all CG beads, as denoted by the CG mapping $M$, are fixed and that no updates are applied to these positions. The complete inference loop is presented in Algorithm 2.

## 2.4. Training Data

**Protein Training Set** The AA training data were collated from previous work (Jones et al., 2023), which consists of over 65K configurations obtained from the Protein Data

---

**Algorithm 2** Inference Loop
---
1: **Input:** Prior noise $\sigma_p$, trained model $EGNN_\theta$, coarse-grained mapping $M$, time step $\Delta t$
2: $x_0 \sim \mathcal{N}(x_1[M], \sigma_p^2\mathbf{I})$
3: $t \leftarrow 0$
4: $x_0[M] \leftarrow \mu_t[M]$
5: **while** $t < 1.0$ **do**
6:     $v_\theta \leftarrow EGNN_\theta(x_t, t, M) - x_t$
7:     $x_t \leftarrow x_t + v_\theta \cdot \Delta t$
8:     $t \leftarrow t + \Delta t$
9: **end while**
10: **return** $x_1$
---

Bank (Berman et al., 2000) via the SidechainNet (King & Koes, 2021) extension of ProteinNet (AlQuraishi, 2019). Sequences between 20-1000 residues in length were included in the training set. All proteins were coarse-grained to representations containing only their $C\alpha$ atoms.

**DNA-Protein Training Set** Sequences were aggregated from the PDB (Berman et al., 2000; 2003) according to all DNA-protein sequences listed in the DNAProDB server (Sagendorf et al., 2017; 2020). Sequences were removed that contained non-cononical base pairs or protein residues. Sequences containing abasic sites or any nucleotide whose atom members deviated from the expected types were also removed. Remaining sequences were stripped of any ions, ligands, waters, or other components that were not DNA or protein chains. More detail on training sets in included in Appendix A2.

## 2.5. Evaluation Metrics

All evaluation metrics have been previously reported and used to compare generatively backmapped structures (Yang & Gómez-Bombarelli, 2023; Jones et al., 2023). Metrics are summarized below, and more detail can be found in Appendix A3.

**Bond Quality Score** evaluates the physical plausibility of generated structures and the bond network by identifying the percent of bonds that are within 10% of the reference bond graph. A bond quality score of 100% is optimal.
**Clash Score** quantifies steric clashes in generated structure by calculating the percent of residues that are within 1.2Åof any other residue. A clash score of 0% is optimal.
**Diversity Score (DIV)** evaluates the consistency of the AA training structure (i.e., ground truth) with the ensemble of generated structures produced by FlowBack. Backmapping is an inherently one-to-many operation, since multiple AA structures are consistent with a single CG structure. It is a desirable feature of a backmapping algorithm to produce a physically plausible diversity of AA structures of which the reference structure is one. A DIV score of 1 indicates all

generated structures are identical (deterministic) and a score near zero indicate high diversity in the generated ensemble.

## 2.6. Model Comparisons

We compare the FlowBack protein model against a rules-based deterministic backmapping approach, PULCHRA (Rotkiewicz & Skolnick, 2008), along with two recently developed generative approaches, GenZProt (Yang & Gómez-Bombarelli, 2023) and DiAMoNDBack (Jones et al., 2023), which are based on a variational autoencoder (VAE) and autoregressive diffusion model, respectively. The GenZProt model has the advantage of fast inference speed as it requires only i) encoding the CG prior ii) sampling from the VAE latent space and iii) decoding latent coordinates into AA structures, however its training set was limited to intrinsically disordered proteins and has shown evidence of mode collapse and limited diversity (Jones et al., 2023). The DiAMoNDBack model achieves improved diversity and clash scores relative to GenZProt, but inference takes relatively longer as each residues must be denoised sequentially. For trajectory data, we make additional comparisons against a DiAMoNDBack model fine-tuned (DiAMoNDBack-FT) on fast-folding protein data from DEShaw Research (DESRES) (Lindorff-Larsen et al., 2011).

## 3. Results

### 3.1. Proteins with all-atom reference

We evaluate our model on two all-atom datasets that have been previously used to benchmark generative backmapping approaches (Jones et al., 2023). In both cases, the model is provided with only the C$\alpha$ trace of each test protein. The first is a set of 24 PDB structures from the CASP12 (Schaarschmidt et al., 2018) challenge consisting of both single and multi-chains and varying from 80 to 600 residues in length. We generated five structures for each protein, corresponding to five unique samples from each CG prior which are integrated forward by our learned ODE. Additionally, we performed inference using a slightly larger Gaussian noise prior than was used during training and term these structure FlowBack-N (details in Appendix A4.1). As illustrated in Table 1, both models outperform previous generative approaches in bond and clash score. Although the vanilla FlowBack model achieves superior performance in these metrics, the higher noise model generated a more diverse ensemble as demonstrated by a diversity score equal to that of the DiAMoNDBack model with very little degradation in the bond and clash performance.

Next we perform inference on coarse-grained traces of 11 fast-folding DESRES proteins each containing 2000 frames(Lindorff-Larsen et al., 2011). In Table 2 we show comparisons to the same models listed above in addition

*Table 1.* Model performance on 24 proteins from CASP12.

| MODEL | BOND (↑) | CLASH (↓) | DIV (↓) |
|---|---|---|---|
| PULCHRA | 98.91 | 0.15 | 1 |
| GENZPROT | 96.26 ±0.01 | 8.43± 0.22 | 0.87 |
| DBACK | 99.18± 0.04 | 0.57± 0.09 | **0.03** |
| FLOWBACK-N | 99.47 ±0.01 | 0.18± 0.25 | **0.03** |
| FLOWBACK | **99.67 ±0.01** | **0.08± 0.09** | 0.19 |

to the DiAMoNDBack-FT model which was fine-tuned on a subset of DESRES data (Jones et al., 2023). FlowBack outperforms all previous models, despite only being trained on PDB structures. Clash score in particular is extremely low with only 0.06% of residues clashing and only 1.4% of structures containing at least one clash. The diversity score is higher than DiAMoNDBack models but lower than GenZProt, and can be improved substantially from 0.23 to 0.08 by using the higher noise variant of the model, again with little degradation in bond and clash performance.

*Table 2.* Model performance across 11 fast-folding proteins generated by DEShaw Research

| MODEL | BOND (↑) | CLASH (↓) | DIV (↓) |
|---|---|---|---|
| PULCHRA | 98.45 | 0.20 | 1 |
| GENZPROT | 94.85 ±0.002 | 6.01± 0.04 | 0.83 |
| DBACK | 97.98± 0.002 | 0.33± 0.01 | 0.03 |
| DBACK-FT | 98.73± 0.004 | 0.18± 0.01 | **0.02** |
| FLOWBACK-N | 98.11 ±0.004 | 0.11± 0.05 | 0.08 |
| FLOWBACK | **99.56 ±0.003** | **0.06± 0.01** | 0.23 |

### 3.2. Coarse-grained protein trajectories

As a final test of our protein model, we backmap trajectories generated by a CG machine-learned forcefield (Majewski et al., 2022) that have no corresponding AA reference structures (details in Appendix A4.1). This task is more challenging as the ensemble of C$\alpha$ positions may be out of distribution compared to PDB training data. As illustrated in Table 3, although the average bond quality for GenZProt and DiAMoNDBack drop on this task, we find that FlowBack continues to perform well as demonstrated by a >99.6% bond for both noise levels. Similarly, clash score increases to >1.3% for comparison models, while clash remains <0.31% for FlowBack. It is not possible to compute a diversity score without a reference structure, however we can use the average RMSD of generated structures with respect to each other as a proxy for this metric (Wang et al., 2022; Jones et al., 2023). We find that the higher noise model can obtain RMSD comparable to that of DiAMoNDBack while still maintaining robust bond and clash scores. There results indicate that our model is generalizable to different forcefields and robust to out-of-distribution CG

traces.

*Table 3.* Average Performance across 3 protein trajectories generated by a CG force-field

| MODEL | BOND (↑) | CLASH (↓) | RMSD (↑) |
|---|---|---|---|
| PULCHRA | 99.23 | 0.75 | 0 |
| GENZPROT | 96.57 ±0.01 | 11.36± 0.03 | 0.21± 0.06 |
| DBACK | 97.73± 0.01 | 1.38± 0.04 | **1.68± 0.21** |
| DBACK-FT | 98.47± 0.02 | 1.26± 0.04 | 1.56± 0.19 |
| FLOWBACK-N | 99.63 ±0.02 | 0.31± 0.10 | **1.69± 0.18** |
| FLOWBACK | **99.71 ±0.01** | **0.25± 0.13** | 1.28± 0.18 |

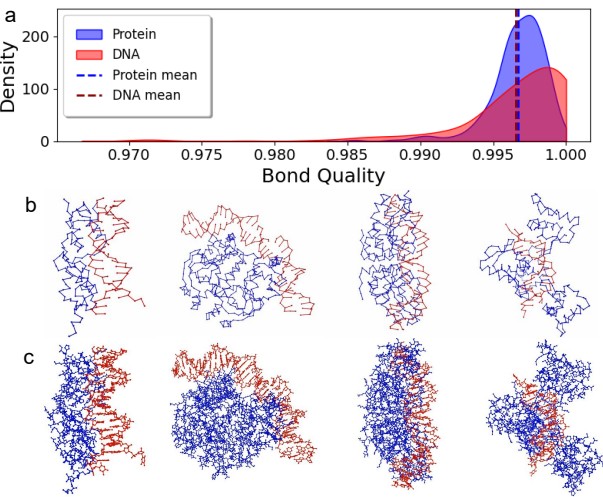

*Figure 2.* a) FlowBack performance on DNA-protein complexes. Bond quality distributions on complexes from the test set. Exemplars of four randomly selected DNA-protein complexes showing the b) CG structures and c) one AA generative backmapping.

### 3.3. DNA-Protein with all-atom reference

We trained a second FlowBack model to reconstruct AA resolution from CG representations of DNA-protein complexes. Proteins were simplified to Cα traces and DNA to 3-site-per-nucleotide (3SPN) configurations (Hinckley et al., 2013). We clustered and reserved 55 structurally distinct complexes for testing and generated five AA structures from the model (details in Appendix A2.5). Evaluation of bond quality in both protein and DNA components against reference PDB structures revealed mean bond accuracies exceeding 99.6%, as illustrated in Figure 2. The bond quality distribution for DNA was broader than that for proteins, with 1% of structures exhibiting bond quality below 99%. The generated structures showed minimal clashes; specifically, 39/275 structures had a single pair of clashing residues, and only 3/275 exhibited multiple clashes (Figure A1). Additionally, the structures maintained substantial populations

of hydrogen bonds, with generated structures containing 89% of protein-protein and 94% of DNA-DNA hydrogen bonds compared to reference structures. However, DNA-protein hydrogen bonds counts were only recovered at 72%, likely due to the structural precision needed for bonding pair formation and the model's tendency to minimize side-chain clashes (Figure A2). These results demonstrate that our model successfully produces realistic structures and interactions.

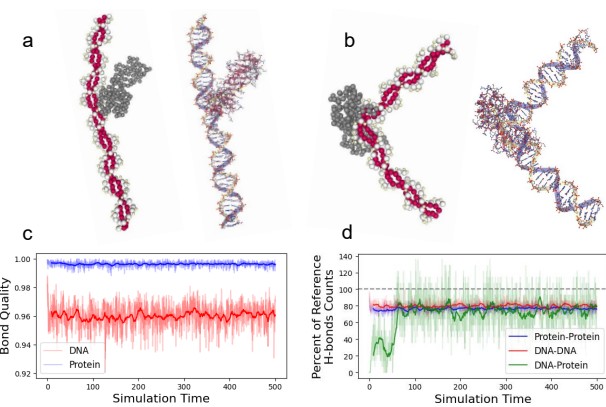

*Figure 3.* FlowBack performance on DNA-protein complex simulations. a-b) CG Frames selected from simulation of TBP and corresponding AA samples generated for those frames. c) Average bond quality of protein and DNA residues over simulation time. d) The percentage of hydrogen bond contacts relative to number of contacts in the corresponding PDB structure (1CDW).

### 3.4. DNA-protein trajectories

Lastly, we applied our DNA-protein model to CG trajectories generated by the AICG-3SPN.2 force-field (Li et al., 2011; Tan et al., 2022) (details in Appendix A2.6). We backmapped 500 frames from simulations perform by Tan et al.(Tan & Takada, 2018) of a TATA-binding protein (TBP) interacting with a 100-base pair DNA strand. CG and AA configurations of the TBP system are shown in Figure 3 along with plots of bond quality and hydrogen bonding over the trajectory. We found that our model achieved excellent bond quality (>99%) for proteins, but produces lower DNA bond score (∼96%) when compared to inference on PDB structures. This is likely due to deformations of the 3SPN.2 structure due to the CG force-field that are not present in training. Still, our models produces physically plausible structures that consistently form hydrogen bonds during different stages of DNA-protein binding and distortion.

### 3.5. Correcting protein stereoisomers

Although the EGNN represents an efficient and expressive network for predicting a conditional vector field, a careful

reviewer pointed out that its lack of reflection invariance can make it prone to predicting incorrect chiralities. Indeed, we found that despite our protein training set consisting of 99.99% L-stereoisomers, 3.7% of predicted residues in the PDB test set were in the D form. Although these may be subtle structural differences, the biological significance of even a small amount of incorrect stereoisomers cannot be ignored. In Appendix A4.4, we include a description of a mechanism that leverages the linearity and robustness of our learned flow in order to ensure all residues are generated in the L-form. Unlike diffusion-based approaches, the flow-matching protocol yields realistic looking structures very early in the ODE integration. As such, we can detect incorrect stereoisomers before all atomic positions have been finalized, reflect the sidechain across the appropriate plane of symmetry, and proceed with the integration given the new structure. We find that this detection and reflection procedure is optimally applied around $t = 0.2$ and has a minimal impact on the bond quality and diversity. However, modifying the structure during the integration process does lead to a slight increase in clash of 0.24% and 0.13% for the PDB and DES test sets, respectively. Updated scores for all evaluation metrics on the PDB and DES test sets are in Appendix A4.4.

## 4. Conclusion

We present FlowBack as a flow-matching approach for generative backmapping of proteins and DNA-protein complexes. We demonstrate state-of-the-art results on structural evaluation metrics as well as tunability to achieve improved diversity during inference. Unlike previous approaches, this architecture and training procedure is not tied to a specific CG map or class of macromolecule, and the paradigm can be generically extended to other classes of molecules such as lipids (Orsi et al., 2008), peptoids (Zhao et al., 2020), or organic semiconductors (Jackson, 2020). Future work may include the exploration of more complex priors beyond Gaussians such as harmonic priors (Stärk et al., 2023; Jing et al., 2023) which leverage information from the bond network to maintain close proximity between neighboring atoms. These priors may be tailored to particular classes of biomolecules or to other data present in the PDB such as small molecules and ions.

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
