# Appendix: FlowBack: A flow-matching approach for generative backmapping of macromolecules

**Michael S. Jones** [1]  **Smayan Khanna** [1]  **Andrew L. Ferguson** [1]

## A1. Backmapping

Given an AA molecular configuration $x^{N_{AA} \times 3}$ we can apply an AA $\rightarrow$ CG map to transform this configuration into $X^{N_{CG} \times 3}$ where $N_{CG} \ll N_{AA}$. It is often not possible to recover the original structure, however we may seek an inverse mapping to deterministically or generatively produce candidate structures $\hat{x}$ that approximate the original configuration.

Given a CG trace and knowledge the AA bond graph, we can construct a "noisy" version $x_0^{N_{AA} \times 3}$ of the target structure $x_1$ by assigning each AA atom to a CG bead according to some mapping $M$ and normally distribute atoms around the bead as $x_0 \sim \mathcal{N}(x_1[M], \sigma_p^2 \mathbf{I})$. The parameter $\sigma_p$ determines how tightly the AA beads are distributed, and the distribution of structures serves as a physically informed prior $q_0(x_0)$ for our flow-matching objective.

## A2. Training and test data

### A2.1. Protein training set

We adopted the same dataset as used previously by Jones et al. (Jones et al., 2023), however we removed all sequences longer than 1000 residues in length. The maximum cutoff of 1000 residues was selected based on the largest sequences that could be accommodated with batch size of 1 in 24 GB VRAM GPU. It should be noted that a much lower VRAM overhead is required during inference as compared to training, therefore excessive VRAM is not required to run the trained model. All structures were previously verified to have a complete set of sidechain and backbone atoms at each residue, physically plausible bond lengths, and no steric clashes or overlaps. A validation set of 100 sequences was randomly selected for hyperparameter tuning. For more details on the data cleaning procedure see (Jones et al., 2023).

### A2.2. Protein test set

We adopted the same test sets as used by Jones et al. (Jones et al., 2023), which includes 24 proteins from the CASP12 protein folding challenge (Moult et al., 2014) and 11 AA trajectories from DEShaw (DES) research (Lindorff-Larsen et al., 2011). The CASP sequences range between 80-600 residues and include 16 single-chain and 8 multi-chain proteins. The DES sequences range from 10-80 and contain 2000 frames evenly strided and concatenated from multiple independent trajectories as decribed previously (Jones et al., 2023).

### A2.3. Coarse-grained protein trajectories

We performed inference on trajectories previously studied by Jones et al(Jones et al., 2023). Trajectories were generated by a machine learned force-field (Majewski et al., 2022) that were parameterized by an equivariant transformer (Thölke & De Fabritiis, 2022) to learn gradients of the energy at each CG bead. We back-mapped 2000 frames of three different types of proteins – BBA, A3D, and PRB – and report the average score across all three proteins in Table 3 of the main text. A comparison of each individual protein to the DiAMoNDBack model is shown below along with visualization of the generated ensemble.

### A2.4. DNA-protein training set

Sequences were aggregated from the PDB (Berman et al., 2000; 2003) according to the list of 5830 sequences contained in the DNAProDB server (Sagendorf et al., 2017; 2020). Sequences were removed that contained non-cononical base pairs or residues. Sequences containing abasic sites or any nucleotide whose atom count deviated from the expected value were also removed. Remaining sequences were stripped of any ions, ligands, waters, or other components that were not DNA or protein chains. A maximum cutoff of 120 DNA base pairs and 500 protein residues was used to filter out extremely large sequences. After completing these filtering steps, a total of 1577 structures remained for training, test, and validation.

[1]UChicago, Pritzker School of Molecular Engineering, University of Chicago, Chicago, Illinois 60637, United States. Correspondence to: Andrew Ferguson <andrewferguson@uchicago.edu>.

*Accepted at the 1st Machine Learning for Life and Material Sciences Workshop at ICML 2024.* Copyright 2024 by the author(s).

### A2.5. DNA-protein test set

We used the MMseqs2 algorithm (Steinegger & Söding, 2017) to build validation and test sets with minimal structural overlap with the train set. We independently clustered DNA and proteins sequences using a coverage of 50% and sensitivity of 0.5, forming 1008 DNA clusters and 455 protein clusters. Because we also test our model on MD trajectories sequences generated from PDBs in our dataset (1pue and 1cdw), we manually placed these sequences in the test set along with six other sequences that shared the same protein or DNA cluster. Next we randomly sampled additional sequences and included all protein and DNA cluster members in the test set until we compiled a minimum of 45 sequences. We repeated this procedure for the validation set, producing a total of 55 sequences in test set and 45 sequences in the validation with no overlap between DNA or protein clusters. Evaluation on the validation set was used for hyperparameter tuning, and results in Figure 2 of the main text are all from the test set. Clash score and diveristy distribution are shown in Figure A1. Hydrogen bond counts are shown in Figure A2.

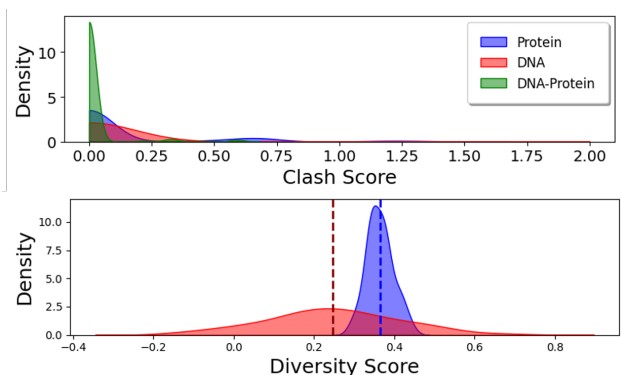

*Figure A1.* Clash and diversity scores computed over five generated samples of all 55 sequences in the protein-DNA test set.

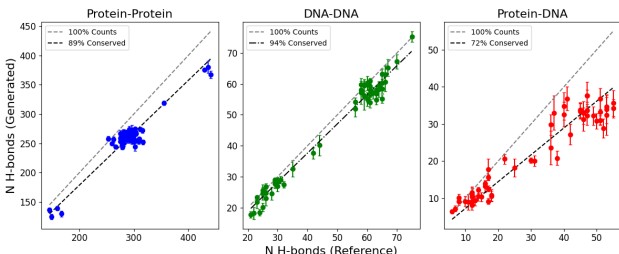

*Figure A2.* Hydrogen bond counts in reference structure vs. average hydrogen bond counts in generated structures. From left to right protein-protein, DNA-DNA, and DNA-proteins complex bond counts.

### A2.6. DNA-protein coarse-grained trajectory

We backmapped CG trajectories kindly provided by the Takada group based on the work of Tan et al. (Tan & Takada, 2018). These trajectories use the 3SPN.2C forcefield (Hinckley et al., 2013; Freeman et al., 2014) to model DNA and the AICG2+ force-field (Li et al., 2011; 2014) for proteins, both of which are compatible with our DNA-protein FlowBack model trained above. An additional position-weight-matrix (PWM) interaction was applied during the simulation to capture the sequence dependent interactions between proteins and DNA. We applied an even striding to obtain 500 frames of a TATA-binding protein (TBP) and generated three samples AA structures per frame. Bond score and hydrogen bond calculations in Figure 3 were performed with respect to a single AA reference structure as there are no AA reference structures for each individual frame.

## A3. Evaluation Metrics

Each molecular configuration and simulation frames was scored by bond score, clash score, and diversity score. Statistics for each score were calculated over 3-5 generated samples per frame. Tables 1-3 in the main text show scores averaged across i) 24 PDB test structures ii) 2000 frames × 11 AA proteins and iii) 2000 frames × 3 CG proteins. Because the diversity score is computed over the generated ensemble, we do not report an error over samples for this metric. The PULCHRA model is deterministic and therefore does have a reportable error for any of the three metrics.

### A3.1. Diversity Metric

As previously defined in Jones et al. (Jones et al., 2023), the diversity metric is calculated by comparing the mean root-mean-squared distance (RMSD) of i) all generated structures with respect to the reference ($RSMD_{ref}$), and ii) the mean RSMD of all generated structures with respect to each other ($RSMD_{gen}$). If these two values are very close to each other, then we obtain a diversity score near zero indicating that the reference structure is indistinguishable from the generated ensemble.

$$RMSD_{ref} = \frac{1}{G} \sum_{i}^{G} RMSD(x_i^{gen}, x^{ref}) \quad (1)$$

$$RMSD_{gen} = \frac{2}{G(G-1)} \sum_{i}^{G} \sum_{j}^{(i-1)} RMSD(x_i^{gen}, x_j^{gen}) \quad (2)$$

$$DIV = 1 - \frac{RMSD_{gen}}{RMSD_{ref}} \quad (3)$$

## A4. Model parameters

### A4.1. Flow-matching parameters

There are three noise parameters that are tuned to optimize flow-matching performance and stability. The standard deviation $\sigma_t$ of the Gaussian used to noise the interpolated structure $\mu_t$ was set to 0.005. The standard deviation $\sigma_p$, used in our prior distribution, was set to 0.003. During inference, the value of $\sigma_p$ can be independently tuned from the value used during training. We achieved the best results for bond and clash scores using a value of 0.003 during inference. However, we achieved better diversity scores—while maintaining strong bond and clash performance—by using a value of 0.005. This later model is reported as FlowBack-N in the main text. Additional bond quality and diversity scores for varied $\sigma_p$ during training and inference are shown in Figure A3. An L1 training objective between the predicted vector field $v_\theta$ and the reference vector field $u_t$ was adopted to enhance stability. We used 100 integration steps for all results presented in the main text.

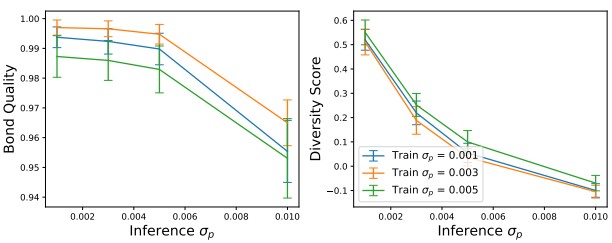

*Figure A3.* Bond score and diversity metric as a function of $\sigma_p$ values used during training and inference.

### A4.2. EGNN Architecture

To fit our model to the flow-matching objective, a neural network architecture is required that can predict a structure $\hat{x}_t$ given some interpolated structure $x_t$ and a corresponding atomic featurization $h$. The model $f$ should be equivariant to rotational and translational transformations $T$ such that the predicted structure is equivalent regardless of the applied transformation $\hat{x}_1 = f(T(x_t), h) = T(f(x_t, h))$. We adopt the Equivariant Graph Neural Network (EGNN) architecture developed by Satorras et al (Satorras et al., 2021) by modifying the implementation available at https://github.com/lucidrains/egnn-pytorch. The model is implemented in PyTorch (Paszke et al., 2019) and uses the Adam optimizer (Kingma & Ba, 2014). We embed a one hot encoding and atom type $a$, residue/bead type $r$, and atom position $p$ and sum these to form our feature embedding. Additionally, we add the flow-matching time $t$ directly to these feature to form our final embedding $h = E_a(a) + E_r(r) + E_p(p) + t$. We did not observe any

benefit by embedding or concatenating the time conditioning. The Euclidean positions of all atomic coordinates were passed in as $x_t$ and the positions of CG beads were masked to ensure their positions would be identical in the output prediction $\hat{x}_t$. Graph edges were specified by the nearest 15 neighbors in Euclidean space. The dimension of the feature embedding and EGNN hidden layers were set to 32. The number of EGNN layers was set to 6; bond quality and clash scores as a function of the number of layers and training epochs are shown in Figure A4. Each training batch only contained one molecular topology (and therefore constant nodes) but included $B$ different times $t$ sampled uniformly on the interval [0, 1]. Batch size was varied on-the-fly as a function of the number of atoms in a given topology $N_a$ and the maximum number of atoms in the training set $N_{max}$ as $B = N_{max}/N_a$. For the protein model $N_{max} = 8070$ and for the DNA-protein $N_{max} = 6299$.

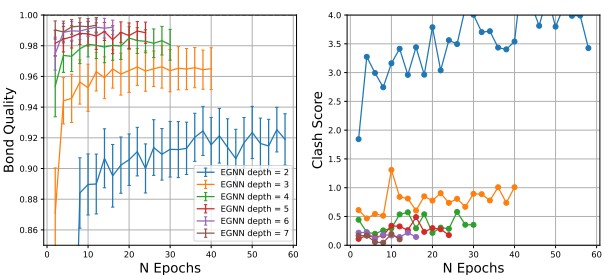

*Figure A4.* Bond and clash score as a function of EGNN layer depth and number of training epochs on the 65k protein training set.

### A4.3. Incorporating DNA residues

Minor modifications to the FlowBack featurization and data-loading procedure were required to accommodate DNA residues. All structures were re-ordered such that proteins residues precede DNA residues, ordered N→C and 5′ → 3′ respectively. We then appended virtual atom positions corresponding to the sugar, base, and phosphate centers of mass used in the 3SPN representation. All DNA atoms were mapped to one of these beads in order to build the CG prior; protein atoms were mapped to their respective C$\alpha$ atoms as described above. The atom type and residue type one-hot encodings were expanded to 68 and 25, respectively, on order to accomodate DNA bead types.

### A4.4. Correcting protein stereoisomers

To detect D-form stereoisomers during the integration process, we pause the ODE at $t_{\text{flip}} = 0.2$ and compute three vectors with respect to each C$\alpha$ for all residues that are not glycines,

$$\mathbf{v}_1 = x_{\mathrm{N}} - x_{\mathrm{C}\alpha}$$
$$\mathbf{v}_2 = x_{\mathrm{C}} - x_{\mathrm{C}\alpha} \qquad (4)$$
$$\mathbf{v}_3 = x_{\mathrm{C}\beta} - x_{\mathrm{C}\alpha}.$$

The chirality with respect to the $C\alpha$ stereocenter can be calculated by finding the signed volume of the parallelepiped formed by these three vectors,

$$V = (\mathbf{v}_1 \times \mathbf{v}_2) \cdot \mathbf{v}_3$$
$$\text{If } V > 0, \quad C_\alpha \text{ is an L center}$$
$$\text{If } V < 0, \quad C_\alpha \text{ is a D center}$$

For all residues that are detected with D center, we apply a reflection of that residue's sidechain atoms across the plane of symmetry formed by $(x_{\mathrm{N}}, x_{\mathrm{C}\alpha}, x_{\mathrm{C}})$. Let $x_s$ be the position vector of an atom in the sidechain. The reflection of $x_s$ across the chiral plane can be computed as follows:

1. Define the normal vector $\mathbf{n}$ to the plane:

$$\mathbf{n} = \frac{(x_{\mathrm{C}_\alpha} - x_{\mathrm{N}}) \times (x_{\mathrm{C}} - x_{\mathrm{N}})}{\|(x_{\mathrm{C}_\alpha} - x_{\mathrm{N}}) \times (x_{\mathrm{C}} - x_{\mathrm{N}})\|}$$

2. Compute the reflection of $x_s$:

$$x'_s = x_s - 2(x_s \cdot \mathbf{n})\mathbf{n}$$

where $x'_s$ is the reflected position vector.

In the case of terminal residues, we apply this reflection to the N or C terminus and do so with respect to the plane defined by $(x_{\mathrm{C}}, x_{\mathrm{C}\alpha}, x_{\mathrm{C}\beta})$ or $(x_{\mathrm{N}}, x_{\mathrm{C}\alpha}, x_{\mathrm{C}\beta})$, respectively.

Although it is possible for the chirality to flip at a later time, we have found that the linearity of the flow tends to lock-in the approximate geometry of sidechain early in the integration process. In order to ensure final configuration retain L-stereocenters, we perform an additional detection and flipping step at $t = 1.0$. The initial detection and flipping time $t_{\mathrm{flip}}$ was identified by sweeping across values from 0.01 to 0.99 and identifying a tradeoff in bond quality and clash score in PDB and DES validation sets (Figure A5). Updated bond quality, clash, and diverity scores with stereocenter correction are shown below for the PDB and DES test sets.

*Table 1.* Model performances on CASP12 test set with and without the addition of chiral (flip) correction.

| MODEL | BOND ($\uparrow$) | CLASH ($\downarrow$) | DIV ($\downarrow$) |
|---|---|---|---|
| FLOWBACK-N | $99.47 \pm 0.01$ | $0.18 \pm 0.25$ | 0.03 |
| FLOWBACK-N (FLIP) | $99.34 \pm 0.01$ | $0.70 \pm 0.13$ | 0.03 |
| FLOWBACK | $99.67 \pm 0.01$ | $0.08 \pm 0.09$ | 0.19 |
| FLOWBACK (FLIP) | $99.57 \pm 0.02$ | $0.32 \pm 0.09$ | 0.19 |

*Table 2.* Model performances on DESRES test set with and without the addition of chiral (flip) correction.

| MODEL | BOND ($\uparrow$) | CLASH ($\downarrow$) | DIV ($\downarrow$) |
|---|---|---|---|
| FLOWBACK-N | $99.11 \pm 0.004$ | $0.11 \pm 0.05$ | 0.08 |
| FLOWBACK-N (FLIP) | $99.15 \pm 0.002$ | $0.45 \pm 0.05$ | 0.07 |
| FLOWBACK | $99.56 \pm 0.003$ | $0.06 \pm 0.01$ | 0.23 |
| FLOWBACK (FLIP) | $99.47 \pm 0.002$ | $0.19 \pm 0.03$ | 0.23 |

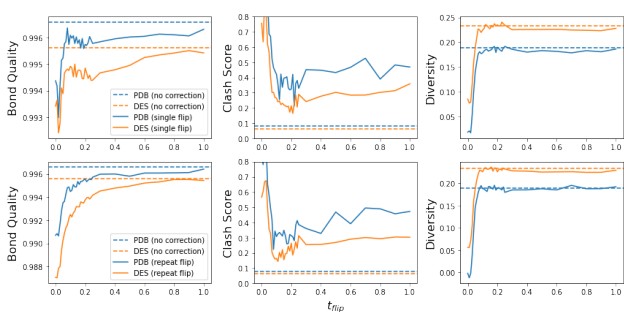

*Figure A5.* Optimizing $t_{\mathrm{flip}}$ in order to maximize the bond quality and clash score of the PDB and DES validation sets. After the specified ODE time, D-form sidechains are identified and flipped once (top row) or continuously (bottom row) in order to produce final structures with desired L-form stereochemistry.