# OpenReview forum: "FlowBack: A Flow-matching Approach for Generative Backmapping of Macromolecules"
_ICML.cc/2024/Workshop/ML4LMS — ML4LMS Oral_

### Official Review · Reviewer_6sRc · 2024-05-30
**A promising flow matching-based approach to generic macromolecule backmapping**

**Rating:** 9
**Confidence:** 4

**Review:**

**Summary:** The authors present a promising flow matching-based approach to the task of generative coarse-grained to all-atom structure mapping.

**Strengths and Weaknesses:**

Points of strength:
- Flow matching for coarse-grained to all-atom structure mapping is quite a promising direction of research in this domain, and the authors have done a great job demonstrating its utility.
- The authors have gone beyond previous works in the literature by training their method to model complex interactions in the Protein Data Bank such as protein-DNA contacts.

Points for improvement:
- I would recommend the authors include additional metrics for proteins besides bond quality and clash scores. For example, using tools such as Molprobity (e.g., for side chain-specific analysis, especially when considering protein-DNA interactions) should add even more insights into the behavior of each baseline method in the authors' experiments.
- The authors may also want to consider whether chirality should be explicitly captured by FlowBack. For example, EGNNs by default are chirality-insensitive (since their scalar node and edge features do not change under global 3D reflections of the input graphs). However, there are many other geometric encoders that directly capture the chirality of the 3D input graphs, so this is something the authors should consider in future work.

**Recommendation:** Given the insightful direction and results of this research and its relevance to this workshop's interests, I highly recommend this work for acceptance.

**Rationale behind Recommendation:** The authors propose a simple yet effective and well designed flow matching model for coarse-grained to all-atom structure mapping, which not only outperforms previous methods based on variational autoencoders and diffusion models but also goes beyond the current literature by training on biomolecules besides proteins (e.g., DNA).

**Questions:**

(1) Do the authors have any intuition for how including ligands (e.g., small molecules, ions, etc) and other types of molecules in the Protein Data Bank might affect which type(s) of prior distributions should be used for generative coarse-grained to all-atom structure mapping? Or should everything be treated with a Gaussian prior? I ask this as an open question for future inquiry.

**Feedback:**
- Vague wording: "on bond quality on clash score"
- Typo: "an CG machine-learned forcefield"
- Typo: "relative to number of contacts (in) the corresponding PDB structure"

**Submission Type:** The authors' submission successfully complies with the corresponding formatting requirements to the best of my knowledge.

---

### Official Review · Reviewer_giAw · 2024-06-05

**Rating:** 7
**Confidence:** 3

**Review:**

This paper studies the problem of protein backmapping, which aims to learn a mapping from coarse-grained models to all-atom data distribution. The paper introduces a simple framework based on flow matching. The method is evaluated on several established becnhmarks to demosntrate the effectiveness.

Generally, the paper studies an important problem with a simple algorithm. However, I still have a question about the experiment evaluation: it seems that in Tables 1&2, there lacks a metric reflecting the "correctness" of the generated structures, so we don't know how the generated structures are close to realistic structures.

---

### Official Review · Reviewer_Pkgt · 2024-06-08

**Rating:** 8
**Confidence:** 4

**Review:**

## Summary

In this work the authors present a flow-matching based for mapping a coarse grained (CG) bio-molecule structure to its potential all atomes (AA) ones.
In particular, they frame this 'super-resolution' problem as a conditional generative models, where the neural network is provided the CG structure (in the form of an additional node features and by 'freezing' the CG atoms).
They empirically show the ability of this model to sample more realistic AA structures (measured in terms of bond quality and steric clashes) both for proteins and NDA.


## Strengths & Weaknesses

The proposed idea is simple and interesting.
Overall the manuscript is pretty well written and the idea well executed.
The presented empirical evidence is pretty convincing.

It might be useful to consider harder tasks where the metrics aren't already saturated.
I believe that it would be worth expending a bit on the benchmarked models, how they differ and why should the proposed approach outperform them.

## Comments

- Are the CG structure representing the Carbon alpha atoms in the protein experiments?
- How DiAMoNDBack differs from the proposed approach?
- I would be curious (yet I acknowledge that requires some extra work) to see how a guidance based approach would perform, as in using an unconditional flow matching model for protein conformation which vector field would be conditioned a posteriori to guide samples to match the provided CG structure.